# Interest of Procalcitonin in ANCA Vasculitides for Differentiation between Flare and Infections

**DOI:** 10.3390/ijms24065557

**Published:** 2023-03-14

**Authors:** Xavier Poirot-Seynaeve, Perrine Smets, Bruno Pereira, Louis Olagne, Julien Stievenart, Vincent Sapin, Olivier Aumaitre, Marc Andre, Ludovic Trefond

**Affiliations:** 1Médecine Interne, CHU Gabriel Montpied, 63000 Clermont-Ferrand, France; 2Biostatistics Unit (DRCI), University Hospital Clermont-Ferrand, 63000 Clermont-Ferrand, France; 3Biochemisty and Molecular Genetic Department, Biology Center, University Hospital Clermont-Ferrand, 63000 Clermont-Ferrand, France; 4M2iSH, UMR 1071 Inserm, INRA USC 2018, University of Clermont Auvergne, 63000 Clermont-Ferrand, France

**Keywords:** ANCA vasculitis, procalcitonin, infection

## Abstract

Procalcitonin (PCT) was established as a biomarker to discriminate bacterial infections from other proinflammatory conditions. Our objective was to determine whether PCT is effective in differentiating infection from antineutrophil-cytoplasmic-antibody (ANCA)-associated vasculitides (AAV) flare. In this retrospective, case-control study, PCT and other inflammatory biomarkers of patients with AAV relapse (relapsing group) were compared to infected AAV patients (infected group). In our population of 74 patients with AAV, PCT was significantly higher in the infected group than in the relapsing group (0.2 µg/L [0.08; 0.935] vs. 0.09 µg/L [0.05; 0.2], *p* < 0.001). Sensitivity and specificity were 53.4% and 73.6%, respectively, for an ideal threshold of 0.2 µg/L. C-reactive protein (CRP) was significantly higher in cases of infection than in relapse (64.7 mg/L [25; 131] vs. 31.5 mg/L, [10.6; 120], *p* = 0.001). Sensitivity and specificity for infections were 94.2% and 11.3%, respectively. Fibrinogen, white blood cell count, eosinophil count, and neutrophil count were not significantly different. In the multivariate analysis, the relative risk of infection was 2 [1.02; 4.5] (*p* = 0.04) for a PCT above 0.2 µg/L. In AAV, PCT may be useful for discriminating between infections and flare in patients suffering from AAVs.

## 1. Introduction

Anti-polynuclear cytoplasmic antibody (ANCA) vasculitides (AAV) are necrotizing systemic vasculitides of unknown cause affecting small vessels. The prognosis of AAV changed with the use of immunosuppressive drugs. In a population of 535 patients with AAV followed between 1995 and 2002, 25% of the patients died within 5 years of diagnosis. In the first year, the main causes of death were infections (48%) and disease activity (19%) [1]. Distinguishing a flare of systemic disease from infection therefore remains a major challenge. In clinical practice, bacterial infection and disease activity can share common symptoms, such as fever, arthralgia, and dyspnea. C-reactive protein (CRP) and white blood cell (WBC) count are not specific enough to distinguish the two conditions [2].

Under normal metabolic conditions, procalcitonin (PCT) is produced by the C-cells of the thyroid gland, where it undergoes proteolysis to produce the hormone calcitonin, which is involved in calcium hemostasis. In the face of pro-inflammatory stimuli, particularly after bacterial infection, it has been shown that several types of cells and organs accelerate the production of PCT [3]. The release of PCT during bacterial infections is directly linked to the presence of certain bacterial constituents (lipopolysaccharides of Gram-negative bacilli) and certain cytokines (TNF, IL-6, IL-1, IL-2) [4]. Its diagnostic properties (sensitivity, specificity) are variable depending on the type of infection. There are false positives, which often correspond to easily identifiable clinical situations, and false negatives, such as localized infection or prior antibiotic therapy. PCT is currently well known and used in infections, especially in intensive care units. PCT was also studied in AAV in small series, although the results were contradictory: in 2015, 10 patients with bacterial infection were compared with 36 patients with AAV flare. With an optimal threshold of 0.1 µg/L, the authors found a sensitivity of 60% and a specificity of 92% for the diagnosis of infection [5]. However, in 2019, in a subgroup analysis of 20 vasculitis patients, the observed PCT levels were not different between the “active vasculitis” (*n* = 16) and “infected” patient (*n* = 4) groups [2]. Currently, we do not have reliable biomarkers for differentiating infection from AAV flare. The objective of this study was to determine whether PCT can be used to differentiate infection from AAV flare. We also analyzed as potential differential factors: temperature above 38 °C, fibrinogen level, and level of various leukocytes.

## 2. Results

### 2.1. Demographic, Clinical, and Laboratory Characteristics of Patients at Diagnosis of AAV

We analyzed 137 patients with an AAV diagnosis. Sixty-three patients without PCT assays were excluded. We finally included 74 patients with a median age of 64 years [51; 73]; 59.4% were men, and there were 43 cases of granulomatosis with polyangiitis (GPA), 21 of eosinophilic GPA (EGPA), and 10 of microscopic polyangiitis (MP). Among these patients, we compared the characteristics of 88 episodes of infection with a PCT assay (infected group) and 72 relapses with a PCT assay (relapsing group). The clinical and laboratory data observed at the time of vasculitis diagnosis are summarized in Table 1.

### 2.2. Characteristics of Relapses and Infections

In the relapsing group, the median corticosteroid therapy was 5.5 mg [5; 10] at relapse. General signs were present in 32 patients (44%), temperatures equal to or greater than 38 °C in 12 patients (16.9%), pulmonary involvement in 37 patients (51.4%), ear–nose–throat (ENT) involvement in 35 patients (48.6%), kidney involvement in 18 patients (25%), ophthalmological involvement in 15 patients (20.8%), cutaneous-articular involvement in 14 patients (19.4%), neurological involvement in 11 patients (15.3%), cardiac involvement in 4 patients (5.5%), and digestive involvement in 4 patients (5.5%). The median Birmingham Vasculitis Activity Score (BVAS) at relapse was 5.5 [3; 10]. Antineutrophil cytoplasmic antibody (ANCA) tests were positive in 62.5% of the cases.

In the infected group, we found 59 bacterial infections (67%), 23 viral infections (26%), and 2 fungal infections (2.3%; 1 pneumocystis and 1 aspergillosis). A temperature equal to or greater than 38 °C was present in 45 infections (51.1%). We had radiological evidence in 37 infections (42%) and microbiological evidence in 32 infections (36.4%). The median dose of corticosteroid therapy was 9 mg [5; 16.4]. We observed 41 bronchopulmonary infections (46.6%), 11 skin infections (12.5%), 10 urinary tract infections (11.4%), 8 ENT infections (9%), 8 gastrointestinal infections (9%), and 7 bacteremia (7.9%), 3 endocarditis (3.4%), and 1 spondylodiscitis cases (1.1%). We had four cases of end-stage renal failure in the relapsing group and seven in the infected group. 

### 2.3. Clinical and Laboratory Markers in the Infected and Relapsing Group

The univariate analysis of the different biological markers and their diagnostic values for the infection are summarized in Table 2. Figure 1 represents an analysis of the values of the different markers between the relapsing group and infected group using boxplots.

The median PCT level was 0.09 µg/L [0.05; 0.2] in the relapsing group and 0.2 µg/L [0.08; 0.935] in the infected group (*p* < 0.001). The median CRP level was 31.5 mg/L [10.6; 120] in the relapsing group and 64.7 mg/L [25; 131] in the infected group (*p* = 0.001).

### 2.4. PCT and CRP Performance for Diagnosis of Infection

The receiver operating characteristic (ROC) curve of PCT and CRP performance is presented in Figure 2. The sensitivity and specificity of PCT for infection were 53.4% and 73.6%, respectively, for an optimal threshold calculated as 0.2 µg/L. The sensitivity and specificity of CRP were 94.2% and 11.3%, respectively, with a threshold of 5 mg/L.

### 2.5. Analysis of the Relative Risk of Infection According to Different Clinical and Biological Parameters

The univariate and multivariate analyses of each biological marker for the diagnosis of infection are summarized in Table 3. In multivariate analysis, a PCT above 0.2 µmol/L was associated with a relative risk of infection of 2 [1.02; 4.5] (*p* = 0.04).

After excluding EGPAs, the median eosinophil level was 0.03 G/L [0.01; 0.09] in the infected group and 0.09 G/L [0.02; 0.24] in the relapsing group (*p* = 0.09). The sensitivity of eosinopenia for infection, with a threshold of 0.03 G/L, was 47.5%, and its specificity was 71.7%.

### 2.6. PCT in Remission and in the Relapsing Group

In the population of 74 patients, we observed 130 PCT assays in the AAV remission period. Median PCT in remission was 0.045 µg/L [0.02; 0.08] versus 0.09 µg/L [0.05; 0.2] in the relapsing group (*p* < 0.001).

## 3. Discussion

In the population of 74 patients with AAV, PCT was significantly higher during infection than during AAV flare. CRP was significantly higher in the infected group than in the relapsing group but had a lower specificity (11.3%) than PCT (73.6%). In the multivariate analysis, the relative risk of infection was 2 [1.02; 4.5] (*p* = 0.04) for PCT above 0.2 µg/L.

In the literature, there are few studies on the interest of PCT in the diagnosis of bacterial infection. High levels of PCT have been observed after proinflammatory stimuli, especially bacterial aggression [6,7]. However, shortly after trauma not involving bacterial aggression (e.g., major surgery, severe burns, newborns at birth, viral infections, allergic disorders, autoimmune diseases, or organ transplant rejection), PCT increased up to only 0.5 µg/L and returned to baseline [8]. Thus, PCT has been established as a biomarker for distinguishing between bacterial infection and other proinflammatory stimuli.

In cases of AAV, contradictory results have been reported [2,5]. Two studies compared PCT levels during infection and AAV flare using another methodology but with only a few patients. Eberhard et al. found a sensitivity of 100% and a specificity of 84% based on 16 infectious episodes in 11 of 35 included patients with AAV (cutoff at 0.5 µg/L). In contrast, the specificity of CRP was less than 15% [9]. Schwenger et al. compared the concentration of CRP in a group of patients with active AAV depending on whether they had a bacterial infection. PCT was higher in the group with infection (median 1.36 µg/L; *n* = 7) than in the group without infection (median 0.19 µg/L; *n* = 17) [10].

In 1998, Moosig et al. compared the PCT levels of 26 patients with AAV during relapse and remission. PCT levels at relapse were below 0.5 µg/L in 23 of 26 patients but were significantly elevated (0.8–3.3 µg/L) in 3 patients [11]. PCT levels were significantly higher at relapse than during remission and were correlated with the disease activity score. In our study, PCT levels were significantly higher at relapse than during remission (0.09 µg/L vs. 0.045 µg/L, *p* < 0.001).

PCT has also been studied in other autoimmune diseases, with controversial results.

Concerning systemic lupus erythematosus (SLE), Shin et al. enrolled 19 patients with fever between 1998 and 1999. Eleven without fever and inactive SLE patients were enrolled as controls. Twelve patients had infection and seven patients had SLE flare. Serum PCT level was significantly higher in patients with infections (0.98 ± 0.12 μg/L) than in those with viral infections (0.13 ± 0.04 μg/L), SLE flare (0.24 ± 0.18 μg/L), or controls (0.12 ± 0.03 μg/L) [12]. In 2013, in SLE patients, Pyo et al. retrospectively recorded 111 febrile episodes [13]. The median PCT level was higher for the 46 infected patients (median, 0.73 µg/L, CI [0.06–6]) than for the 65 patients with febrile flare (median, 0.07 µg/L, CI [0.00–5.67], *p* = 0.002). When PCT levels were above 0.5 µg/L, the relative risk of infection was 11.8 (CI [2.08–66.97], *p* = 0.005). However, this result was not reproduced in a prospective study of 138 patients published in 2019. El-serougy et al. found no difference in the PCT levels of infected versus uninfected SLE patients. In contrast, CRP levels were higher in relapsed patients than in infected patients [14]. In 2012, Sato et al. studied procalcitonin in rheumatoid arthritis (RA). One hundred and eighteen patients with RA flare or nonbacterial infection or bacterial infection were included. All patients with a RA flare had negative PCT levels (≤0.1 ng/mL; *n* = 18). The PCT level was higher in the bacterial infection group (25.8% had levels ≥ 0.5 ng/mL) than in the other three groups (0.0–4.3% had levels ≥ 0.5 ng/mL) and the difference was significant among groups (*p* = 0.003). Conversely, no statistically significant difference was observed among the groups with CRP or WBC. However, PCT < 0.5 ng/mL did not rule out bacterial infection. Here, sensitivity was 14.33% [15].

Outside of autoimmune disease, a threshold of 0.5 µg/L has typically been used [16]. In SLE, the proposed threshold for the diagnosis of bacterial infection ranged from 0.025 to 0.74 µg/L in different studies, with sensitivities ranging from 38% to 89.5% and specificities ranging from 78% to 100% [17,18,19]. For AAV, Ajmani et al. used a PCT threshold of 0.24 µg/L to obtain a sensitivity of 75% and a specificity of 85%. This study included 4 infected patients versus 16 cases of active AAV [2]. In our study, we had a threshold of 0.2 µg/L for a sensitivity of 53.4% and a specificity of 73.6%. This was the best compromise to achieve high specificity while limiting the number of false negatives. These studies did not analyze PCT levels regarding patient renal function.

In a study by Eberhard et al., PCT levels did not differ between patients on dialysis and those with normal renal function [9]. In our study, we did not stratify PCT by renal function, but the influence of renal impairment was not clear. Renal impairment did not alter PCT kinetics in a 2001 pharmacology study [20]. However, elevated levels were observed in patients with chronic renal failure, whether or not they were on dialysis [21,22]. The median PCT levels in patients with stage 5 CKD without substitution were between 0.1 and 1.8 μg/L [23]. The elevated PCT levels in patients with end-stage renal disease were thought to be related to increased proinflammatory mediators in this population. In a 2014 study that included 493 patients admitted to the emergency department or intensive care unit, the ideal PCT threshold for the diagnosis of infection was 1.1 µg/L [24]. Although renal failure may influence PCT levels, in our study, there were no significant differences in the number of cases of end-stage renal diseases between the infected group and the relapsing group.

Our study has some limitations associated with its retrospective design. We used three methods of PCT determination. However, the PCT level did not vary between these methods, according to a multicenter study by Dipalo et al. published in 2015 [25]. Another limitation of our study was the absence of data concerning the possible influence of treatments, particularly corticosteroids, on the determination of PCT. However, in 2008, Perren et al. found comparable levels of PCT in patients with superinfected chronic bronchitis treated with corticosteroids versus a control group without corticosteroids (2.31 vs. 2.13 µg/L, *p* = 0.97) [26], suggesting the absence of influence of corticosteroids on the level of PCT. In our study, for both the infected and relapsing groups, PCT levels were not all performed within the same period after the onset of symptoms. However, PCT tend to be stable in the first few days after an infection according to an in vitro model in a study published in 1994. The half-life of PCT has been estimated to be 24 h [27].

Our study suggests that PCT may be useful for discriminating between infections and flare in patients suffering from AAVs. However, new diagnostic markers are being investigated. The improvement of technologies for biological assays allows, currently, a wide screening of proteins secreted in response to bacterial or viral infection. Other biological markers (presepsin and endocan) could be relevant for differentiating infection from AAV relapse, but they are not currently routinely measured [28,29].

## 4. Materials and Methods

### 4.1. Study Design and Population

We conducted a retrospective study in the internal medicine department of Gabriel Montpied University Hospital in Clermont-Ferrand between 1987 and 2020. Firstly, we included patients who met the 1990 American College of Rheumatology (ACR) criteria of AAV classification [30], the European Medicines Agency algorithm [31], or the Chapel Hill Consensus Conference 2012 definitions for GPA, MP, or EGPA [32]. Secondly, we included patients whose relapses or infections were accompanied by a PCT assay. Thirdly, among the patients with AAV, we identified two groups: (1) an “infected group”, with all infection episodes and a PCT assay, and (2) a “relapsing group,” with all relapses of AAV with a PCT assay.

An event was either an infection or a vasculitis flare. Relapse was defined by a BVAS equal to or greater than 1 [33]. The diagnosis of infection was based on the presence of microbiological evidence. In the absence of bacteriological documentation, an infection diagnosis was made based on a combination of clinical, biological, and compatible imaging abnormalities (conventional radiography or computed tomography scan). Microbiological evidence was based on bacteriological samples such as urinalysis, sputum examination, blood cultures, or the analysis of pleural or peritoneal fluid.

### 4.2. Data Collection

We retrospectively collected demographic, clinical, and laboratory data at the point of AAV diagnosis for each infection and vasculitis flare. For each event, we recorded the temperature and set a threshold equal to or greater than 38 °C as specified in the BVAS classification. We defined general signs according to the BVAS: myalgias and/or arthralgias and/or arthritis or a temperature equal to or greater than 38 °C and/or weight loss greater than 2 kg. Biological markers were measured on patient admission before any antibiotic or immunosuppressive treatment. PCT was also measured during the remission period, which was defined by a BVAS score of 0. CRP was measured by immunonephelometry (VISTA SIEMENS), and fibrinogen was measured by chronometry. CRP levels above 5.0 mg/L and fibrinogen levels above 4.0 g/L were considered positive. PCT was assayed with a two-site sandwich immunoassay using direct chemiluminescence technology with three specific mouse monoclonal antibodies.

Prior to 2003, PCT levels were determined by immunochemiluminescence (LUMItest PCT). The functional sensitivity was 0.1 µg/L. After 2003, PCT levels were determined by immunoluminescence using the Kryptor automaton. The method was fully automated based on time-resolved amplified cryptate emission technology. The functional sensitivity was 0.06 µg/L.

After 2016, PCT was assayed using the ADVIA Centaur^©^ BRAHMS PCT™ assay. ANCA assay was performed by indirect immunofluorescence on human polynuclear smears. The identification of anti-proteinase 3 (PR3) and anti-myeloperoxidase (MPO) was performed by enzyme-linked immunosorbent assay (ELISA). We collected white blood, neutrophil, and eosinophil cell counts. The thresholds for WBC and neutrophil count were 10 G/L and 7 G/L, respectively. Eosinopenia was defined as an eosinophil count below 0.03 G/L.

PCT was also measured during the remission period, which was defined by a BVAS score of 0.

### 4.3. Statistical Analysis

The categorical data were described by frequencies and percentages, while the continuous data were expressed with mean and standard deviation or median and interquartile range, according to their statistical distribution. The assumption of normality (Gaussian distribution) was studied using the Shapiro–Wilk test. First, clinical and laboratory characteristics at AAV diagnosis were described for GPA, EGPA, and MP groups. Then, clinicobiological markers were compared between infected and relapsing groups were compared. Categorical variables (such as temperature ≥ 38 °C) were compared between groups using a chi-squared test or Fisher’s exact test. Continuous variables (i.e., PCT, CRP, fibrinogen, WBC, neutrophils) were compared with Student’s *t*-test or the Mann–Whitney test when the conditions for applying the *t*-test were not met. The homoscedasticity was analyzed using the Fisher–Snedecor test. For PCT and CRP, a ROC curve analysis was then carried out. The most discriminating threshold to predict infected and relapsing groups was determined with regard to clinical relevance and to the indexes usually reported in the literature: Youden, Liu, and efficiency. Diagnostic values (sensitivity and specificity and positive and negative predictive values) were calculated according to these cut-offs for PCT and CRP, whereas thresholds reported in the literature for fibrinogen, WBC, neutrophils, and temperature were used to calculate their diagnostic values. These analyses were then completed by comparisons (univariate and multivariate) between infected and relapsing groups using generalized linear (more precisely logistic) mixed models taking into between and within patient variability (patient as random-effect). Univariate analyses were conducted with all clinicobiological markers treated as categorical variables according to aforementioned thresholds. Multivariate analyses were performed with covariates fixed according to the univariate results and to the clinically relevance. Particular attention was paid to multicollinearity. The final model included the following variables: PCT (≥0.2 µg/L), CRP (≥5 mg/L), and temperature (≥38 °C). Other clinicobiological markers (e.g., WBC ≥ 10,000, neutrophils ≥ 7000 and fibrinogen ≥ 4) were added, step by step, simultaneously, to the aforementioned final model. The results were expressed as relative risks and 95% confidence intervals. All statistical analyses were performed with Stata software (version 15, StataCorp, College Station, TX, USA). Statistical tests were carried out with a two-sided alpha level at 5%.

## Figures and Tables

**Figure 1 ijms-24-05557-f001:**
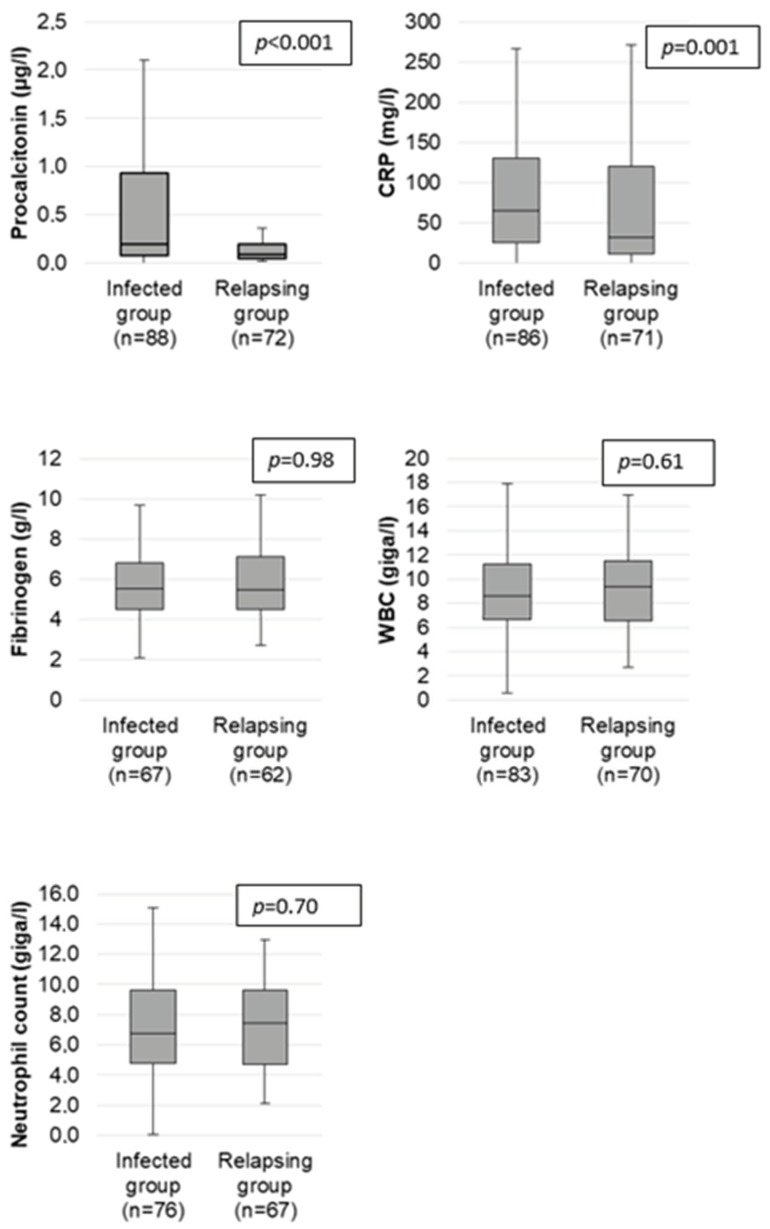
Comparison of PCT, CRP, fibrinogen, leukocytes, and neutrophil levels at relapse and at infection in the population of 74 patients with AAV.

**Figure 2 ijms-24-05557-f002:**
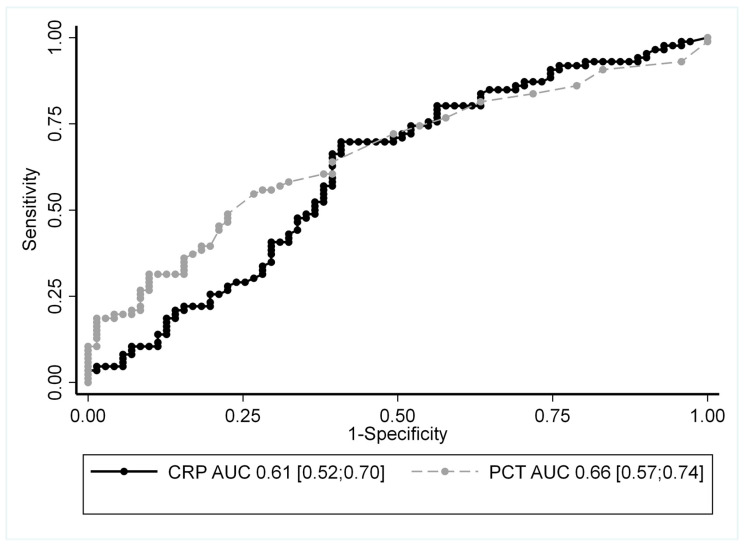
The receiver operating characteristic (ROC) curve of PCT and CRP to differentiate infection from vasculitis flare. AUC: area under the curve and its 95% confidence interval.

**Table 1 ijms-24-05557-t001:** Clinical and laboratory characteristics of the 74 patients at AAV diagnosis.

	Total *n* = 74	GPA *n* = 43	EGPA *n* = 21	MP *n* = 10
**Demographic Data**				
Age at diagnosis (in years) med [Q1; Q3]	64 [51; 73]	64 [50; 75]	59 [51; 64]	79 [72; 82]
Men, *n* (%)	44 (59.4)	23 (53.5)	14 (66.7)	7 (70)
Women, *n* (%)	30 (40.5)	20 (46.5)	7 (33.3)	3 (30)
**Clinical Overview**				
General signs, *n* (%)	59 (79.7)	35 (81.4)	15 (71.4)	9 (79.7)
ENT involvement, *n* (%)	57 (77)	37 (86)	19 (90.5)	1 (10)
Pulmonary involvement, *n* (%)	54 (73)	28 (65.1)	20 (95.2)	6 (60)
Cardiac involvement *n* (%)	12 (16.2)	4 (9.3)	6 (28.6)	2 (20)
Renal involvement *n* (%)	35 (47.3)	26 (60.5)	3 (14.3)	6 (60)
Peripheral neurological disease, *n* (%)	18 (24.3)	10 (13.5)	5 (6.7)	3 (30)
Central neurological disease, *n* (%)	7 (9.46)	4 (9.3)	3 (14.3)	0 (0)
Digestive involvement, *n* (%)	4 (5.41)	3 (6.98)	1 (4.76)	0 (0)
BVAS med [Q1; Q3]	10 (7; 14)	9 (7; 13)	10 (8; 18)	8 (6; 17)
**Diagnosis**				
Histologic evidence, *n* (%)	43 (58.9)	32 (43.8)	6 (28.6)	5 (50)
ANCA, *n* (%)	59 (79.45)	40 (93.02)	9 (40)	10 (100)
IF				
c-ANCA, *n* (%)	32 (43.2)	31 (72.1)	1 (4.76)	0 (0)
p-ANCA, *n* (%)	27 (36.5)	9 (20.1)	8 (38)	10 (100)
Specificity (ELISA)				
PR3, *n* (%)	30 (40.8)	30 (70.7)	0 (0)	0 (0)
MPO, *n* (%)	22 (30.9)	6 (14.6)	6 (30)	10 (100)
**Biological Data Med [Q1; Q3]**				
Creatinine (µmol/L)	76.6 [64; 100] *n* = 73	85 [65; 130] *n* = 43	73.1 [54; 78.9] *n* = 21	68 [64.4; 100] *n* = 9
GFR (mL/min by MDRD)	91 [68; 101.5] *n* = 48	88 [38; 102] *n* = 23	97 [91; 104] *n* = 16	82.3 [61.9; 87] *n* = 9
CRP (mg/L)	83 [19.7; 136]	92.9 [31.7; 147]	29.7 [11.9; 69.8]	132.5 [91; 184]
Fb (g/L)	5.5 [4.8; 7.2] *n* = 70	6.1 [5.2; 7.6] *n* = 39	4.8 [4.2; 5.7] *n* = 21	6.2 [5.2; 7.6] *n* = 10
PCT (µg/L)	0.1 [0.05; 0.3]	0.14 [0.06; 0.39]	0.06 [0.04; 0.14]	0.11 [0.06; 0.32]

Med, median; Q1, first interquartile range; Q3, third interquartile range; GFR, glomerular filtration rate; MDRD, modification of the diet in renal disease; CRP, C-reactive protein; PCT, procalcitonin; BVAS, Birmingham Vasculitis Activity Score; ANCA, antineutrophil cytoplasm antibody; PR3, proteinase 3; MPO, myeloperoxidase; ENT, ear–nose–throat; IF, immunofluorescence; c-ANCA, cytoplasmic antineutrophil cytoplasm antibody; p-ANCA, peri-nuclear antineutrophil cytoplasm antibody.

**Table 2 ijms-24-05557-t002:** Comparison in univariate analysis of clinicobiological markers at relapse or infection and their performance for the diagnosis of infection in 74 patients with AAV.

Markers	Univariate Analysis			Diagnostic Values					
	Relapsing Group *n* = 72	Infected Group *n* = 88	*p*	AUC	Threshold	Se	Sp	PPV	NPV
PCT med (µg/L) [Q1; Q3]	0.09 [005; 0.2]	0.2 [0.08; 0.93]	<0.001	0.66	0.2	53.4%	73.6%	71.2%	56.4%
CRP med (mg/L) [Q1; Q3]	31.5 [11; 120]	64.7 [25; 131]	0.001	0.61	5	94.2%	11.3%	56.3%	61.5%
Fibrinogen (g/L) [Q1; Q3]	5.5 [4.5; 7.1]	5.5 [4.5; 6.8]	0.98	NE	4	89.6%	11.3%	52.2%	50.0%
WBC (G/L) [Q1; Q3]	9.3 [6.6; 11.5]	8.6 [6.7; 11.2]	0.61	NE	10	36.1%	57.1%	50.0%	43.0%
Neutrophils (G/L) [Q1; Q3]	7.4 [4.7; 9.6]	6.8 [4.8; 9.6]	0.70	NE	7	44.7%	46.3%	48.6%	42.5%
T° ≥ 38° n (%)	12 (16.6)	45 (51.1)	<0.001	NE	NE	51.1%	83.1%	78.9%	57.8%

Med, median; Q1, first interquartile; Q3, third interquartile; T°, temperature; AUC, area under the curve; Se, sensitivity; Sp, specificity; PPV, positive predictive value; NPV, negative predictive value; WBC, white blood cell count. NE, not estimated.

**Table 3 ijms-24-05557-t003:** Univariate and multivariate analyses of the relative risk of infection according to different clinical and biological parameters in the population of 74 patients with AAV.

Markers	Univariate Analysis		Multivariate Analysis	
	RR [Q1; Q3]	*p*	RR [Q1; Q3]	*p*
PCT ≥ 0.2 µg/L	1.63 [1.24; 2.15]	**0.001**	2.14 [1.02; 4.50]	**0.04**
CRP ≥ 5 mg/L	1.46 [0.72; 2.95]	0.29	0.96 [0.28; 3.24]	**0.94**
Temperature ≥ 38 °C	1.80 [1.43; 2.43]	**<0.001**	4.16 [1.87; 9.24]	**<0.001**
WBC ≥ 10,000	0.88 [0.64; 1.19]	0.41	0.72 [0.35; 1.49]	0.38
Neutrophils ≥ 7000	0.84 [0.62; 1.15]	0.29	0.65 [0.31; 1.34]	0.24
Fibrinogen ≥ 4	1.04 [0.60; 1.81]	0.88	0.80 [0.22; 2.91]	0.74

RR, relative risk; CI, confidence interval; Q1, first interquartile range; Q3, third interquartile range; BVAS, Birmingham Vasculitis Activity Score; CRP, C-reactive protein; PCT, procalcitonin; WBC, white blood cell count.

## Data Availability

Not applicable.

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
