# Peer review of "Interest of Procalcitonin in ANCA Vasculitides for Differentiation between Flare and Infections"

_ijms, 2023, doi:10.3390/ijms24065557_

Round 1

Reviewer 1 Report

The results of this study will be helpful in medical practice  

Author Response

.

Reviewer 2 Report

The search of biomarkers in vasculitis has been a long lasting journey. The commonly used biomarkers to evaluate infection can not discriminate between infection and disease activity. The evaluation of procalcitonin as a laboratory test that can differentiate infection from vasculitis relapse is challenging.

The presentation of the results could be further improved. 

The methodology to select patient should be described step-step. 

if the authors wish to present data on procalcitonin levels in remission and relapse groups they should elaborate further. 

In multivariate analysis, it should be described whether BVAS score and ANCA were included. 

Reviewer 3 Report

Dear Authors,

The objective was to determine whether procalcitonin (PCT) could differentiate infection from from antineutrophil cytoplasmic antibody (ANCA) associated vasculitis (AAV) flare. The authors presented the results of a retrospective assessment of PCT and other biomarkers of inflammation in two groups of AAV patients. The manuscript should be carefully revised to improve its readability and the relevance of the results. Extensive editing in English is required as the manuscript contains many language errors.

The Introduction section should explain some of the mechanisms associated with PCT levels that can be used to confirm the presence of infection.

The structure of the Results section should be improved.

Were there differences in the parameters between the groups presented in Table 1? How important was the type of vasculitis in assessing PCT and other inflammatory biomarkers?

The text (lines 88-90) repeats the information provided in Table 2.

The supplementary material has of low informative, repeats the information in the maintext.

It is not clear from the legend in Figure 1 where AUC is represented.

The methodology should be more clearly presented. Data collection was carried out between 1987 and 2020 using various equipment. How the data obtained on different equipment were combined? The statistics plan should be more clearly presented.

Please check the abbreviations, as they are not always used correctly.

My overall comment, the manuscript is not ready for publication in its present form, as it has flaws and needs to be edited.

Round 2

Reviewer 3 Report

Dear Authors,

I confirm, that the manuscript has been sufficiently improved to warrant publication in IJMS